# Progression of Pulmonary Function and Correlation with Survival Following Stereotactic Body Radiotherapy of Central and Ultracentral Lung Tumors

**DOI:** 10.3390/cancers12102862

**Published:** 2020-10-05

**Authors:** Sebastian Regnery, Tanja Eichkorn, Fabian Weykamp, Thomas Held, Lisa-Antonia Dinges, Fabian Schunn, Hauke Winter, Michael Thomas, Jürgen Debus, Rami A. El Shafie, Sebastian Adeberg, Juliane Hörner-Rieber

**Affiliations:** 1Department of Radiation Oncology, Heidelberg University Hospital, Im Neuenheimer Feld 400, 69120 Heidelberg, Germany; sebastian.regnery@med.uni-heidelberg.de (S.R.); tanja.eichkorn@med.uni-heidelberg.de (T.E.); fabian.weykamp@med.uni-heidelberg.de (F.W.); thomas.held@med.uni-heidelberg.de (T.H.); lisa-antonia.dinges@med.uni-heidelberg.de (L.-A.D.); fabian.schunn@med.uni-heidelberg.de (F.S.); juergen.debus@med.uni-heidelberg.de (J.D.); rami.elshafie@med.uni-heidelberg.de (R.A.E.S.); sebastian.adeberg@med.uni-heidelberg.de (S.A.); 2National Center for Radiation Oncology (NCRO), Heidelberg Institute for Radiation Oncology (HIRO), Im Neuenheimer Feld 400, 69120 Heidelberg, Germany; 3National Center for Tumor diseases (NCT), 69120 Heidelberg, Germany; hauke.winter@med.uni-heidelberg.de (H.W.); michael.thomas@med.uni-heidelberg.de (M.T.); 4Department of Thoracic Surgery, Thoraxklinik at Heidelberg University Hospital, Roentgenstrasse 1, 69126 Heidelberg, Germany; 5Translational Lung Research Center Heidelberg (TLRC-H), Member of the German Center for Lung Research (DZL), 69120 Heidelberg, Germany; 6Department of Thoracic Oncology, Thoraxklinik at Heidelberg University Hospital, Roentgenstrasse 1, 69126 Heidelberg, Germany; 7Heidelberg Ion-Beam Therapy Center (HIT), Department of Radiation Oncology, Heidelberg University Hospital, 69120 Heidelberg, Germany; 8Clinical Cooperation Unit Radiation Oncology, German Cancer Research Center (DKFZ), 69120 Heidelberg, Germany

**Keywords:** SBRT, non-small cell lung cancer, oligometastases, central, ultracentral, pulmonary function, lung volume, toxicity

## Abstract

**Simple Summary:**

Stereotactic body radiotherapy (SBRT) enables highly focused irradiation of lung tumors and has become a standard treatment. However, SBRT of lung tumors with close proximity to the central airways or mediastinum (central and ultracentral tumors) is associated with an increased risk for severe complications (bronchial bleeding, blockage of bronchi with loss of lung function). This retrospective study analyzed lung function and survival after risk-adapted approaches of SBRT in 107 central and ultracentral lung tumors. Lung function (vital capacity, forced expiratory volume in the first second) showed a statistically significant but in absolute numbers modest decrease that correlated moderately with the maximum radiation dose to the central airways. Stronger decrease in pulmonary function was found to be associated with limited survival. Consequently, lung function tests should be an integral element of follow-up after SBRT of lung tumors with proximity to the central airways or mediastinum.

**Abstract:**

Stereotactic body radiotherapy (SBRT) to central and ultracentral lung tumors carries a risk of excessive toxicity. This study analyzed changes in pulmonary function tests (PFT) and their correlation with overall survival (OS) in 107 patients following central (*n* = 62) or ultracentral (*n* = 45) lung SBRT. Ultracentral location was defined as planning target volume overlap with the proximal bronchial tree (PBT). Vital capacity (VC) (−0.3 l, absolute −9.4% of predicted, both *p* < 0.001) and forced expiratory volume in the first second (FEV_1s_) (−0.2 l, absolute −7.7% of predicted, both *p* < 0.001) significantly decreased following SBRT. Higher maximum dose to the PBT significantly correlated with a steeper decline in VC (*p* = 0.005) and FEV_1s_ (*p* = 0.03) over time. Pronounced decline in FEV_1s_ between 6 and 12 months (HR = 0.90, *p* = 0.006) and pronounced decline in VC between baseline and 12 months (HR = 0.95, *p* = 0.004) independently correlated with worse OS. Consequently, PFT presented a statistically significant albeit clinically mild decrease in lung volumes following central and ultracentral SBRT that correlated moderately with maximum dose to the PBT. Stronger decline in pulmonary function was associated with constrained survival, advocating consequent performance of PFT during follow-up.

## 1. Introduction

In the past few years, stereotactic body radiotherapy (SBRT) has advanced to a standard treatment in medically inoperable patients with early-stage non-small cell lung cancer (NSCLC) [1,2,3] as well as pulmonary oligometastases [4,5]. Nevertheless, SBRT of lung tumors in a central location, defined as a location less than 2 cm from the proximal bronchial tree (PBT) [6], remains a challenge. Application of sufficiently high doses to the tumor must be weighed against the risk for possibly severe toxicity [7,8,9]. In recent years, it has been demonstrated that the risk for excessive toxicity and even mortality increases with proximity to the PBT, so that ultracentral tumors in contact with the PBT are at especially high risk [7,8,10]. While more protracted SBRT fractionations have been successfully established for treatment of central tumors [2,6], there is still no strong evidence or consensus on treatment for ultracentral lesions [9]. Besides the increased incidence of bronchopulmonary bleeding [8,11], bronchial strictures and loss of lung volume are among the major concerns following central and ultracentral SBRT [7]. However, data on pulmonary function testing following central and especially ultracentral SBRT are scarce. Most analyses incorporate only peripheral tumors [12,13] or a low number of central tumors [14,15]. Only one retrospective analysis directly compared toxicity due to decrease in PFT parameters between central and peripheral lung tumors [16]. Data on ultracentral lesions are lacking. The aim of this study is to investigate the time course of pulmonary function test (PFT) parameters following risk-adapted SBRT of central and ultracentral lung tumors.

## 2. Results

### 2.1. Pulmonary Function Parameters

The simple course of PFT parameters over time was evaluated descriptively as well as employing linear mixed models (LMM) with time as a fixed effect and a random intercept for each subject to deal with longitudinal data structure. The vital capacity (VC) presented a statistically significant decline in absolute −0.3 l (β = −0.2 l per 6 months, *p* < 0.001) and absolute −9.4% of the predicted VC (β = −5.1% per 6 months, *p* < 0.001) within 12 months post-SBRT. Similarly, forced expiratory volume in the first second (FEV_1s_) significantly decreased by absolute −0.2 l (β = −0.1 l per 6 months, *p* < 0.001) and by absolute −7.7% of the predicted FEV_1s_ (β = −4.4% per 6 months, *p* < 0.001) within 12 months post-SBRT. The ratio VC/FEV_1s_ did not present a time trend in the descriptive analysis, which was confirmed by a statistically non-significant β-coefficient in the LMM (β = 0.1% per 6 months, *p* = 0.89). Results of the simple time course analysis are summarized in Table 1 and illustrated in Figure 1.

VC and FEV_1s_ given as percentages of the predicted values were further assessed employing LMM with time, one additional clinical or dosimetry variable and the respective interaction term as fixed effects. Details of the LMM assessment are given in Table 2. 

The VC showed a statistically significant positive correlation with the mean biologically effective dose (with an assumed α/β ratio = 3, BED_3_) in both lungs (β = 14.2% per 10 Gy) and a positive correlation with planning target volume (PTV) size (β = 0.6% per 10 cm^3^). Moreover, the maximum BED_3_ to the PBT (β = −0.6 per 6 months and 10 Gy) and the mean BED_3_ in both lungs (β = −4.8% per 6 months and 10 Gy) yielded statistically significant negative interactions with time. Assessment of all significant variables in a joint LMM showed time (β = 6.7% per 6 months), mean BED_3_ in both lungs (β = 10.3% per 10 Gy), the interaction between time and the maximum BED_3_ to the PBT (β = −0.5% per 6 months and 10 Gy) and the interaction between time and the mean BED_3_ in both lungs (β = −4.6% per 6 months and 10 Gy) as statistically significant predictors of the VC.

The FEV_1s_ significantly correlated with the mean BED_3_ in both lungs (β = 21.0% per 10 Gy), PTV size (β = 0.6% per 10 cm^3^) and the Charlson Comorbidity Index (CCI) (β = −3.1% per 1 point). A statistically significant interaction with time was shown for maximum BED_3_ to the PBT (β = −0.4% per 6 months and 10 Gy). Joint analysis of all significant variables showed CCI (β = −2.8% per 1 point), mean BED_3_ in both lungs (β = 13.6 per 10 Gy) and the interaction between time and maximum BED_3_ to the PBT (β = −0.4 per 6 months and 10 Gy) as statistically significant predictors of the FEV_1s_.

Figure 2 and Figure 3 illustrate the course of VC and FEV_1s_ over time for selected patient subgroups. Appendix A present further patient subgroups. Appendix A yields the course of FEV_1s_/VC for different subgroups over time, which were only analyzed descriptively.

### 2.2. Overall Survival

Univariate regression yielded decreased baseline FEV_1s_/VC ratio (hazard ratio (HR) = 0.98), decreased baseline FEV_1s_ as percentage of predicted FEV_1s_ (HR = 0.99), stronger decline in FEV_1s_ between 6 and 12 months (FEV_1s_ as percent of predicted: HR = 0.91, absolute FEV_1s_ in liter: HR = 0.02), stronger decline in VC between baseline and 12 months (VC as percent of predicted: HR = 0.97, absolute VC in liter: HR = 0.35) as well as stronger decline in VC between 6 and 12 months (VC as percent of predicted: HR = 0.95, absolute FEV_1s_ in liter: HR = 0.30) as statistically significant predictors of worse OS. Multivariate regression showed an independent association of stronger decline in FEV_1s_ between 6 and 12 months (FEV_1s_ as percent of predicted: HR = 0.90) and stronger decline in VC between baseline and 12 months (VC as percent of predicted: HR = 0.95) with poor OS. Table 3 summarizes all results from Cox regression analysis.

## 3. Discussion

To our knowledge, we analyzed the largest dataset of PFT following central and ultracentral lung SBRT as dedicated subgroups so far. Both VC and FEV_1s_ significantly declined following SBRT, with a mean reduction of absolute −9.4% in VC and absolute −7.7% in FEV_1s_ expressed as percentage of the predicted value. Given baseline values of 85.6% for VC and 69.5% for FEV_1s_, this corresponds to an average relative decline of slightly more than 10%. Hence, the average decline reaches a PFT-related toxicity grade I according to the RTOG 0813 scale [6,16]. Accordingly, SBRT is confirmed as a safe treatment for central and ultracentral lung tumors considering changes in pulmonary function. However, we found a significant correlation of dynamic PFT changes with OS. Therefore, some patients might be at higher risk due to more pronounced PFT changes. This advocates for the regular use of PFT during follow-up after central and ultracentral SBRT to detect vulnerable patients early. Even though the linear models suggest a continuous decline, descriptive analysis reveals that the decline in PFT is most pronounced during the first 6 months post-SBRT, with smaller changes towards the 12-month follow-up. This suggests the time points we investigated as very reasonable candidates for follow-up studies.

### 3.1. Central vs. Peripheral SBRT

In general, several previous studies of PFT following SBRT could not find significant changes in FEV_1s_ and/or VC [13,15,17,18]. Conversely, Stanic et al. found a significant relative −5.8% change in FEV_1s_ two years after peripheral SBRT [19]. Accordingly, Stone et al. reported a relative drop in VC of around 4.6% 12 months after SBRT in one of the largest cohorts of peripheral lung tumors reported so far [12]. These numbers suggest a smaller decline than the one we observed. Hörner-Rieber et al. showed a significant decline in FEV_1s_ by absolute 5.2% at a median of 9 months after SBRT, including a small number of central tumors (*n* = 12/70) [14]. Guckenberger et al. investigated PFT values following SBRT in a patient cohort potentially including central tumors, finding a significant relative −8.1% change in FEV_1s_ between 7 and 24 months [20]. Both studies agree well with our findings, which also suggest that tumor location might have an impact on PFT changes following SBRT. While we could not find a significant correlation of ultracentral versus central tumor location with PFT changes, there was a significant interaction between decline in PFT and the maximum dose delivered to the PBT. Consequently, higher radiation doses to the PBT could indeed have an adverse impact on pulmonary function after SBRT. In this context, Stephans et al. compared a small subgroup of central tumors (*n* = 10) with peripheral tumors without finding significant PFT changes after SBRT in either group [15]. Furthermore, Mangona compared toxicity after peripheral and central SBRT in a larger collective, showing similar ≥II° PFT toxicity rates of around 35% after 2 years for both groups [16]. Notably, such toxicity rates also suggest significant changes in PFT after SBRT in general. All in all, analyses of sufficiently large patient cohorts show that SBRT induces statistically significant but clinically moderate decreases in pulmonary function. Despite a potentially pronounced decline following higher radiation doses to the PBT, our data do not raise concerns about excessive toxicity due to loss of lung volume following risk-adapted central or ultracentral SBRT. The decline in FEV_1s_ of absolute −0.2 l or relative −11.8% still compares favorably to PFT changes following surgical resection. A decline in FEV_1s_ of −0.4 l or relative −17.6% depending on the extent of the resection was reported in a prior study comparing PFT after pulmonary surgery and SBRT [21]. Moreover, patients undergoing surgery are generally younger and present with less comorbidity [21].

### 3.2. Clinical and Dosimetry Correlation

SBRT is a standard treatment for patients with early-stage lung cancer unfit to undergo surgical resection or for patients with pulmonary oligometastases. Hence, interpretation of PFT changes over time following SBRT might be biased due to advanced age, severe comorbidities, tumor progression or consecutive systemic treatments. Consequently, we expanded our LMM to encompass factors other than SBRT which are possibly associated with a decline in PFT over time. The CCI showed a negative correlation with the FEV_1s_ in the final linear model, whereas patient age, local tumor progression, application of consecutive systemic treatments or incidence of pneumonitis did not show such a correlation in any of the employed models. The correlation of CCI with FEV_1s_ suggests that FEV_1s_ is generally lower in patients with higher burden of comorbidities. However, the interaction term between CCI and time did not show a significant effect. This establishes SBRT as an important reason for pulmonary function changes during the follow-up period, which is further supported by the stronger decline in PFT during the first 6 months after SBRT, as was visible in the descriptive analysis.

Due to the application of an internal target volume (ITV) concept for motion management, the mean PTV size of 105 cm³ was somewhat elevated compared to prior reports of lung SBRT [14,17], which might have contributed to a more pronounced decline in pulmonary function. Nevertheless, declines in pulmonary function remained modest.

Higher mean dose in both lungs was associated with significantly higher VC and FEV_1s_ in general, which suggests that patients who received higher lung doses also started at higher baseline PFT values. Potentially, SBRT dose was chosen more aggressively to improve tumor control in patients with better pulmonary function. Prior studies showed a weak [15] or no correlation of PFT decrease with higher lung doses [14,20]. Accordingly, interaction terms between mean BED_3_ in both lungs and time after SBRT showed a statistically significant but clinically moderate correlation with VC. In particular, an additional 10 Gy mean BED_3_ to both lungs (corresponding to an excess of two standard deviations in our patient cohort) theoretically translated to a further decline in VC of absolute −4.6% per 6 months.

Furthermore, LMM suggested that increased maximum BED_3_ to the PBT was associated with a significantly pronounced decline in PFT. However, clinical effect size was limited, with incremental changes of absolute −0.5% and −0.4% per 10 Gy BED_3_ and 6 months for VC and FEV_1s_, respectively. 

Even though advanced patient age, a high burden of comorbidities and tumor progression surely contributed to the observed decrease in PFT parameters over time, our results suggest SBRT as one major reason for PFT decline. Despite being weak to moderate, correlations with mean lung dose as well as with maximum dose to the PBT support a risk-adapted approach to fractionation and dosing in central and ultracentral SBRT.

### 3.3. Overall Survival

Pre-treatment pulmonary function did not yield independent predictors of OS on multivariate analysis. While one prior study failed to show a correlation of pre-treatment PFT with overall survival [19], many authors found a correlation of more favorable pre-SBRT PFT with decreased survival [12,15,17]. This counterintuitive finding could be explained by severe cardiovascular disease as the main reason for medical inoperability in patients with good PFT in the corresponding studies, subsequently leading to higher cardiovascular mortality [12,15,17]. We found a stronger decline in VC between baseline and 12 months post-SBRT as well as a stronger decline in FEV_1s_ between 6 months and 12 months post-SBRT to be independent predictors of poor OS. A previous analysis of the association between post-treatment PFT parameters and OS could not show a correlation of poor PFT parameters with poor OS [12]. However, we did not analyze singular post-treatment PFT values, but the dynamic changes in PFT compared to baseline pulmonary function. Similarly, a previous analysis from our department including mainly peripheral tumors showed that the actual decline in the VC was an independent predictor of worse OS [14]. The progression of predicted FEV_1s_ between 6 and 12 months had the highest impact on OS, yielding a hazard ratio of 0.90 per 1% absolute increase or 1.11 per 1% absolute decrease. This is intriguing because FEV_1s_ seems to reach a plateau between 6- and 12-months post-treatment in descriptive analysis. Obviously, a further decline in FEV_1s_ in this timeframe confers an increased risk of mortality. Considering our results from correlation of PFT with clinical parameters, neither higher age nor higher burden of comorbidities significantly interacted with the PFT decrease following SBRT. Therefore, some patients might develop a stronger decrease in PFT independently of age and previous comorbidities. Consequently, our results advocate for the inclusion of PFT in regular follow-up visits after central or ultracentral lung SBRT. This might enable early detection of pronounced PFT changes as a predictor of worse outcome.

### 3.4. Limitations and Strenghts

Strengths of this study include conduction of all PFT at the same center. Furthermore, we report the largest cohort of patients with central and especially ultracentral tumors followed up with PFT after SBRT so far. All patients received standardized risk-adapted SBRT fractionations depending on tumor location. Besides this, our analysis suffers from several general limitations due to its retrospective nature. Firstly, investigation of a heterogeneous patient cohort that includes mostly elderly and frail patients or patients with advanced tumor disease limits interpretation of PFT changes over time. To adjust for some important confounders, we elaborated various LMM with patient age, comorbidity status and following systemic therapies as additional covariates. Other possible confounders such as smoking status, pulmonary comorbidities in specific as well as out-field tumor progression were not included in our analysis. Follow-up data on PFT were not available for all patients, which might have confounded our results. Moreover, PFT were performed at time points slightly varying from the ideal baseline, 6- and 12-months post-treatment. Additionally, spirometry relies on the patient’s effort to cooperate during the examination, which could have distorted the longitudinal measurements.

## 4. Materials and Methods 

### 4.1. Patients

This retrospective analysis included 129 patients who consecutively received SBRT of central or ultracentral lung tumors at Heidelberg University Hospital between 2012 and 2019. Of these, 22 patients were excluded due to missing pre-treatment (baseline) PFT, so that a total of 107 patients were eligible for analysis. Central location was defined according to the RTOG 0813 trial [6] and ultracentral location was defined as an overlap of the PTV with the PBT [10]. Two patients received SBRT to two central lung lesions at the same time and only the bigger PTV was included in statistical modeling. All available data on pre-treatment comorbidities were used to retrospectively assign a CCI to each patient. The clinical outcomes as well as dosimetry parameters of this patient cohort are currently submitted for publication. Patient characteristics are shown in Table 4. 

### 4.2. Radiation Treatment

Planning of SBRT employed a 4-dimensional (4D) CT imaging set. Patient immobilization relied on individually shaped body casts. To mitigate tumor motion, an abdominal compression device was employed if the tumor was in a lower lung lobe. Contouring of gross tumor volumes (GTVs) on several maximum extension phase images (encompassing maximum inspiration and expiration, middle breathing position) was performed to create an ITV. Clinical target volumes (CTV) were obtained by adding a 2–5 mm margin to the ITV. PTVs were obtained by adding another 3 mm margin to the CTV. Normal tissue constraints were in line with the UK consensus guidelines [22], with heart, lungs, PBT, esophagus, chest wall and spinal cord being regularly delineated as organs at risk (OAR). Due to central and ultracentral tumor location, risk-adapted SBRT fractionations were employed. Patients with central lung tumors were treated to a total dose of 60 Gy in 8 fractions prescribed to the encompassing 80% isodose [2]. Ultracentral lung tumors were treated to a total dose of 50 Gy in 10 fractions prescribed to the encompassing 95% isodose. OAR constraints were given priority over target coverage. Delivery techniques were either 3D, volumetric-modulated arc therapy (VMAT) or helical tomotherapy. Simultaneous systemic therapy was not performed. To enable comparability between the two different SBRT fractionations, the doses to lungs and PBT were converted to the biologically effective dose assuming an α/β ration of 3 (BED_3_) according to the well-known linear-quadratic model: BED_α/β_ = n × d(1 + d/(α/β))

### 4.3. Follow-Up

Patients were followed up every 3–6 months after SBRT, including clinical evaluation, plain radiography or computed tomography (CT) of the thorax and PFT at our institution. Most PFT data were available around 6 months and around 12 months post-SBRT. Regularly, PFT was based on spirometry and entailed VC, FEV_1s_ as well as the ratio of FEV_1s_/VC (Tiffeneau index). VC and FEV_1s_ were available as absolute values in liter (l). Moreover, corresponding expected lung volumes were predicted based on patient age, sex and body height according to the global lung function 2012 equations [23], which enabled expression of the different PFT parameters as percent of the predicted value (% predicted). Local progression was defined as increase in tumor size within the high dose volume according to RECIST 1.1 on thoracic CT scan (*n* = 9). Furthermore, several patients received additional positron emission tomography (PET) CT (*n* = 2) or biopsy (*n* = 4) to confirm local progression. Occurrence of pneumonitis grade ≥II° was retrospectively registered based on clinical and imaging findings according to the common terminology criteria of adverse events (CTCAE) in version 5.0. Overall survival (OS) was calculated from the first day of SBRT to patient death.

### 4.4. Statistics

Patient as well as tumor characteristics were evaluated descriptively. Furthermore, the development of PFT over time was analyzed descriptively for the whole patient cohort as well as between different patient subgroups. For quantitative assessment of PFT changes over time, LMM were developed for each PFT parameter as target variable. Due to longitudinal data structure, the individual subject was included as random effect so that all models encompassed one random intercept per patient. Firstly, basic time models were built with follow-up time (discretized to baseline, 6 months, 12 months) as the only fixed effect, showing a significant correlation of FEV_1s_ as well as VC with follow-up time. For further evaluation, only FEV_1s_ and VC values given as percent of the predicted volumes were chosen to reduce bias by age and sex. One more fixed effect and its interaction term with time were added to the basic time models of FEV_1s_ and VC to evaluate possible differences in time trend. Lastly, all fixed effects that proved at least borderline significant (p ≤ 0.05) were incorporated into a final LMM for FEV_1s_ and VC. Variances were estimated based on the restricted maximum likelihood (REML) method. Confidence intervals were computed as profile confidence intervals and p-values were derived from the z-statistics. Q-Q-plots of the residuals as well as random intercepts for all LMM can be found in Appendix A, showing approximately normal distributions for all models. Potential correlation of OS with baseline PFT as well as PFT differences between baseline and the two follow-ups was investigated using univariate Cox proportional hazard models. All variables that were statistically significant in univariate analysis were incorporated into a multivariate model. If both absolute values (in l) as well as normalized values (expressed as percent of predicted) proved statistically significant, only the corresponding normalized value (percent of predicted) was chosen for the multivariate model to avoid intercorrelation between the predictor variables. Confidence intervals and *p*-values of all HR were derived from Wald statistics. Level of statistical significance was set to α < 0.05. Since this was an exploratory analysis, *p*-values were calculated without adjusting for multiple comparisons and should be regarded as descriptive in nature. Statistical analysis was conducted with R in version 3.6.0 using the “rspiro” and “lme4” package.

### 4.5. Ethics

This retrospective trial received approval by the local ethics board (IRB number: S226/2020) and was conducted in accordance with the Declaration of Helsinki. Individual patient data will not be made publicly available according to national legislation and study ethics approval terms.

## 5. Conclusions

In general, SBRT of central and ultracentral lung tumors led to statistically significant albeit clinically mild changes in pulmonary function that were correlated moderately with the mean lung dose and maximum dose to the proximal bronchial tree. Hence, risk-adapted central and ultracentral SBRT is supported as a safe treatment modality. A stronger decline in pulmonary function was associated with constrained survival, which advocates for regular PFT follow-up in the first year after SBRT.

## Figures and Tables

**Figure 1 cancers-12-02862-f001:**
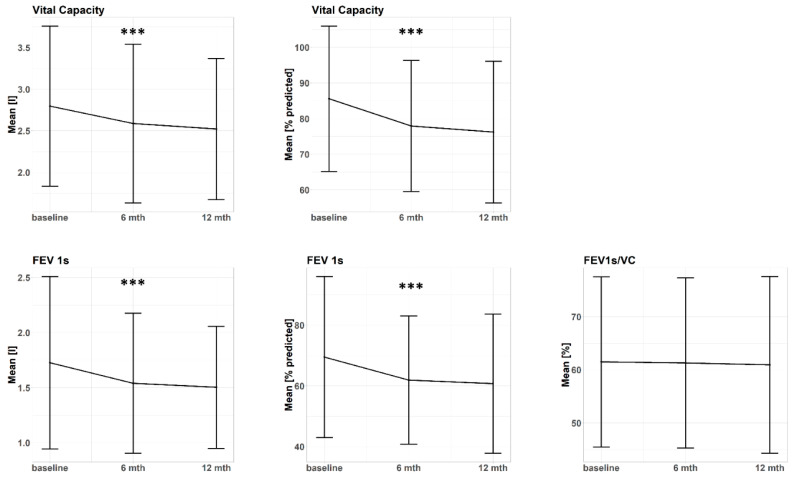
Progression of different pulmonary function parameters following central or ultracentral stereotactic body radiotherapy (SBRT). Data points represent mean values and error bars represent ± 1 standard deviation. FEV 1s: forced expiratory volume in the first second, VC: vital capacity, l: liter, mth: months, ***: *p* < 0.001.

**Figure 2 cancers-12-02862-f002:**
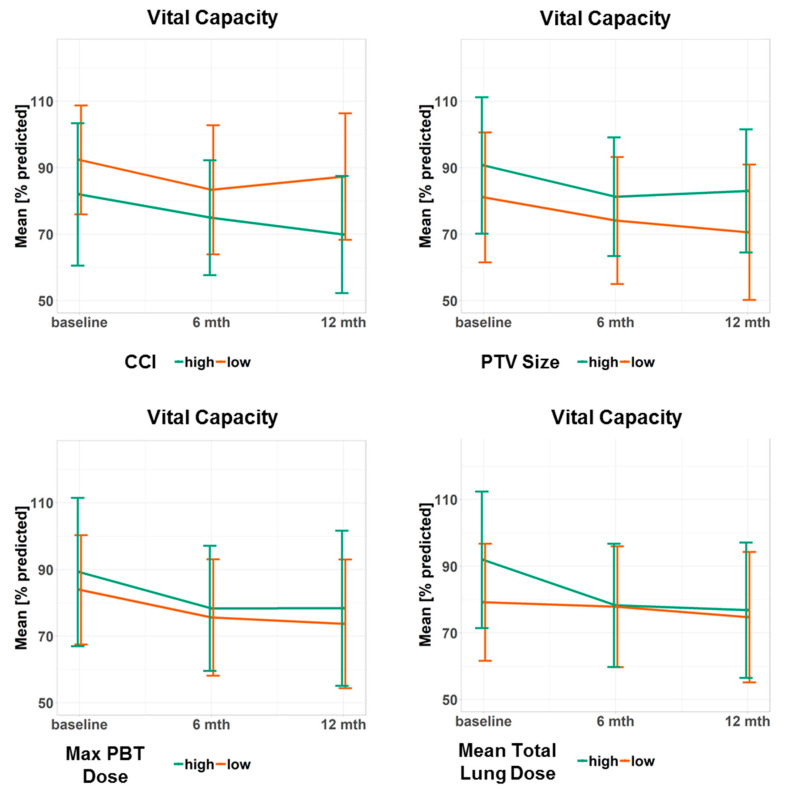
Progression of the vital capacity for different subgroups. Data points represent the mean value at a given time point. Error bars visualize ± 1 standard deviation. The continuous predictor variables were dichotomized at the median to obtain two groups, one with high and one with low values of the variable. Doses refer to the biologically effective dose based on an α/β ratio = 3. CCI: Charlson Comorbidity Index, PTV: planning target volume, PBT: proximal bronchial tree, max: maximum.

**Figure 3 cancers-12-02862-f003:**
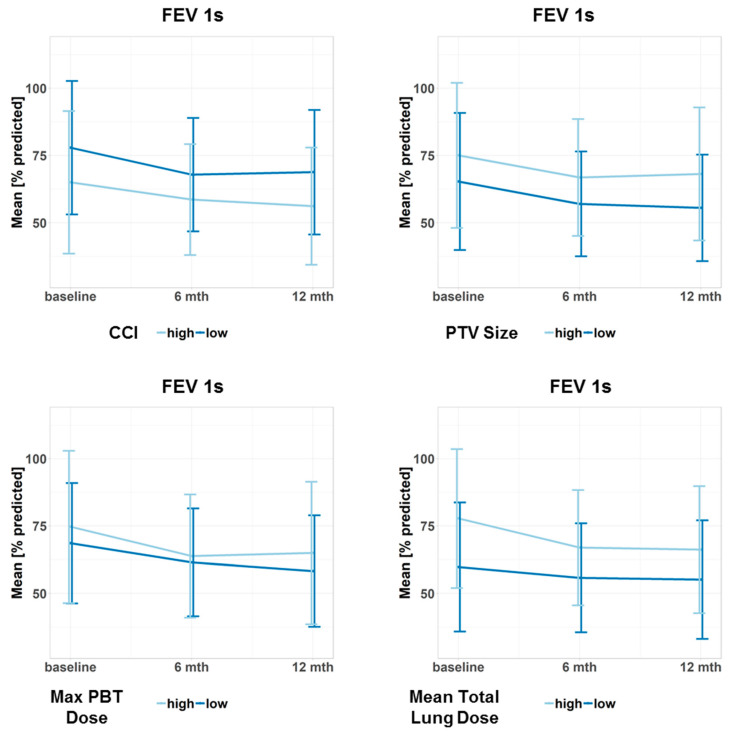
Progression of the forced expiratory volume in the first second (FEV 1s) for different subgroups. Data points represent the mean value at a given time point. Error bars visualize ± 1 standard deviation. The continuous predictor variables were dichotomized at the median to obtain two groups, one with high and one with low values of the variable. Doses refer to the biologically effective dose based on an α/β ratio = 3. CCI: Charlson Comorbidity Index, PTV: planning target volume, PBT: proximal bronchial tree, max: maximum.

**Table 1 cancers-12-02862-t001:** Changes in different pulmonary function tests over time. Values are given as mean ± standard deviation. Linear mixed model-based estimates for the β-coefficients of time as fixed effect are given with 95% confidence intervals (CI) and corresponding p-values. VC: vital capacity, FEV_1s_: forced expiratory volume in 1 second, Δ: absolute difference.

Pulmonary _Function Parameter	Baseline (*n* = 107)	6 Months (*n* = 73)	12 Months (*n* = 55)
FEV_1s_ [l] (Δ baseline)	1.7 ± 0.8 (0)	1.5 ± 0.6 (−0.2 ± 0.3)	1.5 ± 0.6 (−0.2 ± 0.3)
β estimate [l/6 months]	β = −0.1 [−0.2–−0.1], *p* = 2 × 10^−9^		
FEV_1s_ [% predicted] (Δ baseline)	69.5 ± 26.5 (0)	61.9 ± 21.1 (−8.5 ± 12.6)	60.8 ± 22.9 (−7.7 ± 12.4)
β estimate [%/6 months]	β = −4.4 [−6.0–−2.9], *p* = 1 × 10^−8^		
VC [l] (Δ baseline)	2.8 ± 1.0 (0)	2.6 ± 1.0 (−0.2 ± 0.5)	2.5 ± 0.9 (−0.3 ± 0.5)
β estimate [l/6 months]	β = −0.2 [−0.2–−0.1], *p* = 6 × 10^−8^		
VC [% predicted] (Δ baseline)	85.6 ± 20.4 (0)	77.6 ± 18.4 (−7.8 ± 14.9)	76.3 ± 19.9 (−9.4 ± 14.5)
β estimate [%/6 months]	β = −5.1 [−6.9–−3.3], p = 2 × 10^−8^		
FEV_1s_/VC [%] (Δ baseline)	61.5 ± 16 (0)	61.3 ± 16.0 (−0.9 ± 6.1)	60.9 ± 16.6 (0.5 ± 6.8)
β estimate [l%/6 months]	β = 0.1 [−0.7–0.8], *p* = 0.89		

**Table 2 cancers-12-02862-t002:** Linear mixed model analysis. All models include one random intercept per individual subject (random effect). Reference values for each fixed effect are included in brackets. Estimates for the β-coefficients of the fixed effects are given with 95% confidence intervals (CI) and corresponding p-values. VC: vital capacity, FEV_1s_: forced expiratory volume in 1 second, CCI: Charlson Comorbidity Index, ST = systemic treatment, BEDα/β: biologically effective dose based on the α/β ratio, PBT: proximal bronchial tree, PTV = planning target volume, •: interaction term.

Fixed Effects	VC [% Predicted]		FEV_1s_ [% Predicted]	
	β [95% CI]	*p*-Value	β [95% CI]	*p*-Value
Age (1 year)	0 [−0.4–0.4]	0.98	0 [−0.5–0.4]	0.91
Time (6 months)	−14.2 [−26.4–−1.9]	0.02	−9.8 [−23.4–−2.4]	0.02
Age Time	0.1 [0–0.3]	0.14	0.1 [0–0.3]	0.11
CCI (1 point)	−1.5 [−3.6–0.6]	0.15	−3.1 [−5.7–−0.6]	0.02
Time (6 months)	−4.9 [−9.6–0.2]	0.04	−5.7 [−10.3–−2.4]	2 × 10^−3^
CCI Time	0 [−1.1–1.0]	0.93	0.2 [−0.4–1.3]	0.31
Localization (central)	6.0 [−1.4–13.5]	0.11	7.5 [−1.8–16.8]	0.12
Time (6 months)	−5.3 [−7.6–−3.0]	6 × 10^−6^	−4.5 [−6.4–−2.5]	1 × 10^−5^
Localization • Time	0.6 [−3.0–4.2]	0.75	−1.1 [−3.0–3.1]	0.97
Following ST (none)	−0.3 [−10.4–9.8]	0.95	6.3 [−6.3–18.9]	0.33
Time (6 months)	−5.0 [−7.1–−2.8]	5 × 10^−6^	−4.0 [−5.8–−2.2]	2 × 10^−5^
Following ST Time	−0.3[−5.0–4.5]	0.91	−1.3 [−5.6–2.5]	0.45
Pneumonitis (none)	3.8 [−7.0–14.5]	0.49	6.8 [−17.8–15.8]	0.32
Time (6 months)	−5.1 [−7.1–−3.2]	1 × 10^−7^	−4.4 [−6.1–−2.9]	1 × 10^−7^
Pneumonitis Time	0.4 [−4.9–5.7]	0.88	−0.2[−4.1–5.0]	0.94
Local Progression (none)	5.1 [−8.3–18.6]	0.46	−1.0 [−20.5–15.8]	0.91
Time (6 months)	−5.1 [−7.0–−3.2]	2 × 10^−7^	−4.5 [−6.6–−3.1]	7 × 10^−8^
Local Progression Time	−0.3 [−5.6–5.0]	0.88	0.4 [−3.3–6.7]	0.85
PBT Max (10 Gy)	0.4 [−0.4–1.2]	0.33	0.1 [−0.9–1.1]	0.82
Time (6 months)	2.6 [−2.7–7.9]	0.34	0.5 [−4.2–5.2]	0.82
PBT Max Time	−0.6 [−1.0–−0.2]	3 × 10^−3^	−0.4 [−0.7–0]	0.03
Total Lung Mean (10 Gy)	14.2 [5.7−22.6]	1 × 10^−3^	21.0 [10.6–31.4]	8 × 10^−5^
Time (6 months)	−0.2 [−5.0–4.7]	0.95	−1.1 [−5.2–3.0]	0.61
Total Lung Mean Time	−4.8 [−9.2–−0.4]	0.03	−3.1 [−6.8–0.6]	0.10
PTV Size (10 cm³)	0.6 [0.1–1.0]	0.01	0.6 [0.1–1.2]	0.03
Time (6 months)	−3.5 [−6.4–−0.7]	0.02	−3.6 [−6.1–−1.1]	5 × 10^−3^
PTV Time	−0.1 [−0.4–0.1]	0.16	−0.1 [−0.3–0.1]	0.31
Time (6 months)	6.7 [0.1–13.4]	0.05	0.6 [−4.1–5.3]	0.80
CCI (1 point)	−−−	−−−	−2.8 [−5.3–−0.4]	0.02
PBT Max Time (10 Gy • 6 months)	−0.5 [−0.9–−0.2]	5 × 10^−3^	−0.4 [−0.7–0]	0.03
PTV Size (10 cm³)	0.3 [−0.2–0.8]	0.18	0.2 [−0.4–0.9]	0.44
Total Lung Mean (10 Gy)	10.3 [0.4–20.1]	0.04	13.6 [2.0–5.2]	0.02
Total Lung Mean Time (10 Gy • 6 months)	−4.6 [−8.9–−0.2]	0.04	−−−	−−−

**Table 3 cancers-12-02862-t003:** Analysis of overall survival. Correlation of overall survival with different parameters of pulmonary function tests over time. Regression analysis was based on Cox proportional hazard models. HR: hazard ratio, CI: confidence interval, FEV_1s_: forced expiratory volume in 1 second, VC: vital capacity, pred: predicted value, Δ: absolute difference, BEDα/β: biologically effective dose based on the α/β ratio, PBT: proximal bronchial tree.

Univariate		
Pulmonary Function Parameter	HR (95% CI)	*p*-Value
FEV_1s_ [l]	0.93 (0.64–1.37)	0.73
FEV_1s_ [% pred]	0.99 (0.98–1.0)	0.03
VC [l]	1.22 (0.92–1.62)	0.18
VC [% pred]	0.99 (0.98–1.01)	0.34
FEV_1s_/VC [%]	0.98 (0.96–1.0)	0.01
Δ FEV_1s_ (0–6 months) [l]	1.25 (0.39–3.97)	0.71
Δ FEV_1s_ (0–6 months) [% pred]	1.02 (0.99–1.04)	0.29
Δ FEV_1s_ (0–12 months) [l]	0.40 (0.13–1.26)	0.12
Δ FEV_1s_ (0–12 months) [% pred]	0.99 (0.96–1.01)	0.30
Δ FEV_1s_ (6–12 months) [l]	0.02 (0–0.21)	1 × 10^-3^
Δ FEV_1s_ (6–12 months) [% pred]	0.91 (0.86–0.97)	4 × 10^-3^
Δ VC (0–6 months) [l]	0.951 (0.48–1.87)	0.88
Δ VC (0–6 months) [% pred]	1.0 (0.98–1.02)	0.81
Δ VC (0–12 months) [l]	0.35 (0.17–0.74)	6 × 10^−3^
Δ VC (0–12 months) [%pred]	0.97 (0.95–1.0)	0.03
Δ VC (6–12 months) [l]	0.30 (0.15–0.63)	1 × 10^−3^
Δ VC (6–12 months) [%pred]	0.95 (0.92–0.98)	1 × 10^−3^
Δ FEV_1s_/VC (0–6 months) [%]	1.03 (0.98–1.09)	0.19
Δ FEV_1s_/VC (0–12 months) [%]	1.02 (0.97–1.07)	0.42
Δ FEV_1s_/VC (6–12 months) [%]	1.0 (0.95–1.06)	0.93
Multivariate		
FEV_1s_/VC [%]	1.0 (0.95–1.05)	0.86
FEV_1s_ [% pred]	0.97 (0.94–1.0)	0.09
Δ FEV_1s_ (6–12 months) [% pred]	0.90 (0.84–0.97)	6 × 10^−3^
Δ VC (0–12 months) [% pred]	0.95 (0.91–0.98)	4 × 10^−3^
Δ VC (6–12 months) [% pred]	1.0 (0.95–1.05)	0.94

**Table 4 cancers-12-02862-t004:** Patients (*n* = 107). SD: standard deviation, CCI: Charlson Comorbidity Index, IQR: interquartile range, NSCLC: non-small cell lung carcinoma, PET: positron emission tomography, SBRT: stereotactic body radiotherapy, BEDα/β: biologically effective dose based on the α/β ratio, PBT: proximal bronchial tree, PTV: planning target volume, PFT: pulmonary function test, Δ: difference.

Patient Characteristics	
Age (years)	
Mean ± SD	71.6 ± 10.4
Sex	
Male	52
Female	55
CCI	
Median (IQR)	4 (3–5)
Localization	
Central	62
Ultracentral	45
Tumor Entity	
NSCLC	88
PET Positive Lung Nodule *	12
Extrapulmonary Primary	7
SBRT Target	
Primary	68
Local Recurrence	20
Lung Metastasis	19
Systemic Therapy during Follow-Up	
Chemotherapy	8
Checkpoint Inhibition	6
Chemotherapy + Checkpoint Inhibition	1
Other	4
None	74
Unknown	14
Maximum BED_3_ in PBT	
Mean ± SD	115.7 ± 49.2
Mean BED_3_ in Total Lung	
Mean ± SD	10.1 ± 4.4
PTV Size (cm³)	
Mean ± SD	105.0 ± 84.0
PFT Time Intervals (days)	
Δ Baseline–SBRT Start (Mean ± SD)	−42 ± 30
Δ SBRT Start–1st Follow-Up (Mean ± SD)	182 ± 37
Δ SBRT Start–2nd Follow-Up (Mean ± SD)	359 ± 45
Pneumonitis during Follow-Up	
II°	10
III°	5
Local Progression during Follow-Up	9

* Cases where histological proof was not possible and lung lesion showed (^18^F)-fluorodeoxyglucose (FDG) utilization on positron emission tomography typical for bronchial carcinoma.

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
