# Peer review of "Progression of Pulmonary Function and Correlation with Survival Following Stereotactic Body Radiotherapy of Central and Ultracentral Lung Tumors"

_cancers, 2020, doi:10.3390/cancers12102862_

Round 1
Reviewer 1 Report
Interesting work.
Please use cm3 instead of ccm.
The mean PTV size value is high for SBRT, which I believe is due to the method used for ITV contouring (maximum inspiration and expiration). These things needs a comment in the Discussion.
Please better explain that the “declined pulmonary function was associated with constrained survival”. It seems that the declined pulmonary function is solely due to SBRT. In your series, 88 patients were affected by NSCLC. I think that out-field tumor progression might contribute to decrease pulmonary function as well as overall survival.
Reviewer 2 Report
This manuscript evaluates the time course of pulmonary function test parameters following risk-adapted SBRT of central and ultra central lung tumors. It is a well written paper. There are very limited information about the PFT and SBRT correlation, particularly in central and ultra central lesions.
The analysis is well design. However, is complicated evaluating or looking for possible relationships among OS and PFT. The co-morbilities (like pulmonary diseases) and smoking history were not included in this study.
This article could be improved by including data on smoking history and evaluating the relationship with PFT and OS.
It would be interesting to make explicit reference to the residual pulmonary function after surgery in the introduction (similar or different?).
This study is a useful addition to the literature with the focus on lung cancer SBRT and pulmonary function. And show that, the SBRT is a well tolerated treatment to short- and long-term.
